# Fish Size Correlates to Size and Morphology of Intermuscular Bones in Tambaqui *Colossoma macropomum* as Shown by Dissection and X-ray Imaging Methods

**Naislan F. A. Oliveira** [1], **Lucas S. Torati** [2], **Luciana A. Borin-Carvalho** [1], **Leandro K. F. de Lima** [2], **Velmurugu Puvanendran** [3,*], **Thaís H. Demiciano** [4], **José J. T. da Silva** [2], **Aurisan da S. Barroso** [2] and **Eduardo S. Varela** [2]

1   Centro de Ciências Agrárias, Departamento de Agronomia, Maringá, Universidade Estadual de Maringá-UEM, Maringá 87020-900, PR, Brazil; naislanfernanda@hotmail.com.br (N.F.A.O.); labcarvalho@uem.br (L.A.B.-C.)
2   EMBRAPA Fisheries and Aquaculture, Palmas 77008-900, TO, Brazil; lucas.torati@embrapa.br (L.S.T.); leandro.kanamaru@embrapa.br (L.K.F.d.L.); josejunior.zootecnia@gmail.com (J.J.T.d.S.); aurisan.zootecnista@gmail.com (A.d.S.B.)
3   Center for Marine Aquaculture, Nofima AS, Muninbakken 9, 9019 Tromsø, Norway
4   Complexo de Ciências Agrárias, Departamento de Engenharia Agronômica, Universidade Estadual do Tocantins—UNITINS, Palmas 77020-122, TO, Brazil; thaishelenad7@gmail.com
*   Correspondence: velmurugu.puvanendran@nofima.no; Tel.: +47-77-629-216

**Abstract:** One of the main issues in the processing sector of the tambaqui *Colossoma macropomum* is the removal and/or fragmentation of intermuscular bones (IBs), which negatively impacts its production chain. In this sense, we quantitatively examined the IB variation in farmed tambaqui (n = 127) by comparing the direct anatomical dissection with the high-resolution X-ray imaging method. The number of IBs from the anatomical dissection on the left side of the fish (27.3 ± 5.70 bones) was comparable to that of X-ray analysis (26.9 ± 6.03 bones) ($p > 0.05$). In addition, 76% of deviation in IB number between the two studied methods was one to three, indicating both methods are equally efficient for identifying and quantifying IBs. We found a strong positive correlation (R = 0.8, $p < 0.001$) between the X-ray and the dissection methods. Our predictive models indicated that more than 50% of variation in IB length can be explained by growth parameters. Our results demonstrated that the X-ray method can provide accurate phenotypic data (in vivo) for IB counting and length measurements by extrapolating from the standard length, body weight and trunk over axis area of tambaqui.

**Keywords:** intermuscular bones; tambaqui; X-ray

**Key Contribution:** Our study highlights that variation in intermuscular bones in tambaqui exists. Ours is the first study comparing the intermuscular bone number morphology by dissection and X-ray imaging methods in tambaqui. Our study confirms that intermuscular bone number and morphology can be characterized non-lethally by using X-ray imaging.

## 1. Introduction

The global human population is rapidly increasing, and the demand for additional food to feed the increasing human population is huge. Having higher omega-3 fatty acids, fish has been promoted as a superior protein-rich nutritious food compared to other animal products, and this presumably changed the eating habits of humans [1]. Fish consumption has doubled in the last few decades and is forecasted to increase another 100% in the next two decades due to global demand for a protein source for an ever increasing human population [2]. However, meeting the increased demand for protein and omega-3-rich fishery products cannot be achieved by capture fisheries because they have seen almost zero growth in the last few decades, which means fish from aquaculture are the only choice

to meet this demand [3]. In Brazil, per capita fish consumption is low compared to the global consumption (less than 10 kg compared to 20 kg, respectively); nevertheless, it has been increasing in the last two decades, which follows the similar global trend [4]. Seafood consumption in Brazil is higher than its own production; thus, Brazil has huge potential to increase its seafood production due to its 8400 km marine coastline and larger volume of freshwater resources, comprising 12% of the global freshwater resources [5]. Globally, carps, catfish and shrimps in Asia, salmonids, breams and bass in Europe, salmonids and catfish in North America and salmonids, shellfish species and barramundi in Oceania are the major fish and invertebrate species farmed [6]. So far, there are five major farmed aquatic species in Brazil—tambaqui, tilapia, white leg shrimp, arapaima and catfish—but tambaqui and arapaima are specific to Brazil and other South American countries [4,7].

The tambaqui *Colossoma macropomum* (Cuvier, 1816) is the second largest scaled fish after the pirarucu (*Arapaima gigas*) and the second most farmed fish species in Brazil after the tilapia (*Oreochromis niloticus*) [7,8]. Among the native species, tambaqui leads the production rank with 18.2% of the 551,900 tons of fish produced in 2020 [9]. In Brazil, tambaqui production predominates in the northern region, where it represents 73.0% of the total 100,600 tons of the species farmed in the country [9]. Tambaqui constitutes the major protein source for the people living in various Amazonian regions, which resulted in intense fishing efforts which led to this species being over-exploited [10,11]. Being a biologically resilient fish species that endures extremes of oxygen, temperature and pH, tambaqui has been considered to have huge potential for farming in Brazil [7,12]. The relative easiness for acquisition of healthy juveniles, the satisfactory growth potential and the meat acceptance by the consumer market are the main attractions for the aquaculture of tambaqui in the country [13]. As such, production of juveniles has also gained momentum due to the decrease in natural stocks of the species and the higher quality of products from aquaculture over fisheries [11,14]. The fishery and aquaculture production of tambaqui has increased from 8 tons in 1994 to 139,000 tons in 2014, and farming alone contributed 102,600 tons [15]. The basic spawning, larval rearing and broodstock conditions are now known, and currently, tambaqui is the largest farmed teleost behind tilapia due to versatility of adaptation to various extensive, semi-intensive and intensive farming conditions [15]. With desirable characteristics such as taste, attractive white colour and presence of a reasonable amount of fat and high protein content, tambaqui meat is appreciated both in the national and international markets, thus consolidating its social and economic importance [7,16].

Intermuscular "Y" bones (IBs), which are slender bones embedded in muscles, are uniquely found in the myosepta of teleostean fishes [17]. The number of IBs in fish varies depending on the species, while cyprinoid fishes are well known for having these IBs [18]. The function of the IBs is correlated with morphological metrics and swimming [19]. However, they present some choking hazard in humans during consumption, which provides some negative impacts on consumer preference and marketing [20]. While some fish do not need extensive post-harvest processing, other fish species, such as rohu (*Labeo rohita*; [21]), several species of Asian carps [22–24] and hilsa (*Tenualosailisha*; [25]), require processing before marketing due to the presence of intermuscular "Y" bones (IBs), which deter the consumers from readily accepting it [26]. Similarly, in *C. Macropomum*, consumer preference and marketing is limited by the presence of IBs, which forbid the diversification of cuts demanded by consumers, such as strip, rib, loin and fillet, without bones [14]. IBs are small spicule-like bones existing in the muscle fillet, specifically in the myosepta on both sides of the vertebrae [27]. Presence of IBs is considered a key bottleneck for tambaqui industry expansion in Brazil [27]. The mechanical removal of the "Y" bones after such cuts is not very well accepted by the industry, since mechanical removal causes a loss in fillet yield in addition to being time consuming, resulting in a more expensive final product [28,29].

Recently, a captive population of *C. macropomum* lacking intermuscular bones has been identified, which suggests the existence of significant phenotypic variation. This variation has the potential to be used as a trait in selective breeding programs of tambaqui aiming at either reduction in or elimination of these skeletal structures [27,30]. Such

achievements would represent a breakthrough for the tambaqui farming industry, since it would open novel markets for novel processed products, thus increasing production and aggregated value [14]. However, so far there is limited knowledge on the IB variations in *C. macropomum*, especially during early development. Furthermore, there is no validation of a method for in vivo evaluation of IB morphology and number. Our study is relevant in terms of providing a non-invasive in vivo method for the evaluation of IB number and morphology to develop techniques and equipment for processing and for future selective breeding of the species targeting reduction or elimination of IBs [14,31]. Dissection is a more accurate method because all the IBs can be retrieved, but the fish needs to be sacrificed. To measure the traits needed for selective breeding, the measurement procedures need to be simple and non-lethal to keep the selected individuals alive to produce the next-generation progeny. So, this work aimed to make a comparative analysis of direct anatomical dissection and high-resolution X-ray imaging methods for diagnosis and quantification of tambaqui IBs. Use of high-resolution X-ray images improved the quality of the images obtained, [32] and later, we validated the accuracy of X-ray imaging in comparison with dissection for counting IBs, as well as observing presence, absence, types, lengths and distribution of the IBs in different body areas of tambaqui.

## 2. Materials and Methods

### 2.1. Sampling

For this study, we used 127 juveniles that were 229-days-old *C. macropomum*, averaging $18.55 \pm 2.21$ cm in standard length (SL), from the same family that were produced at the research station of Embrapa Fisheries and Aquaculture (Palmas, TO, Brazil). These individuals had been implanted intramuscularly at the fingerling stage with a Passive Integrated Transponder (PIT-tag) (Marca Pet, Morretes, PR, Brazil), so X-ray and direct dissection data of the same fish could be compared later. Prior to analysis, specimens were sacrificed with a lethal dose of Eugenol 10% (2–5 mg/L). Their body weight was measured to the nearest 0.0 g, and then they were preserved in 100% ethanol until later analysis.

### 2.2. X-ray Procedures

For X-ray analysis, we followed methods described by Perazza, Menezes, Ferraz, Pinaffi, Silva and Hilsdorf [27], and specimen integrity was maintained after changing ethanol periodically until completion of the analyses. Before analyses, PIT-tags were read and fish were individually positioned in right lateral decubitus to take images, thus referring to fish left side (Figure 1A), in line with our objective of verifying the efficiency of X-ray as a diagnostic method for intermuscular bones examination.

To obtain the images, we used a portable digital X-ray device (JPI Healthcare Solutions, Ronkonkoma, NY, USA, model JPI 9020HF, ExamVue (1.0.30.12)). Operating with a voltage and current of 40–90 kV and 20 mA, respectively, a maximum output of 1.60 kW and 100 kHz was produced. A set of four or six fish per plate were placed in the right lateral decubitus position, and each fish on the plate sets was marked with its origin, after which the X-ray image was taken. The images were generated in the DICOM extension and then converted to JPG, and the photos were cropped using the Windows 10 photo editor program to separate each fish image from the set of four or six fish, making each specimen stay in individual files.

These images were used to evaluate the localization, quantification, length measurement and the morphological aspect of the intermuscular bones and to measure the standard and head length of each fish. Following the methodologies from previous studies, these assessments were made only on the left side, with the fish in the anatomical position of right lateral decubitus [14,31]. After the X-ray, specimens were placed in containers with 70% ethanol until the dissection.

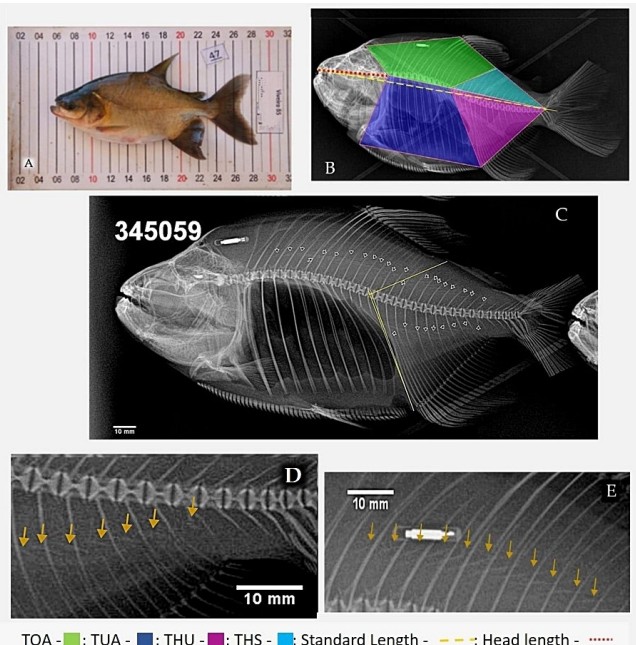

TOA - ▮; TUA - ▮; THU - ▮; THS - ▮; Standard Length - – – –; Head length - ······

**Figure 1.** (**A**)—Tambaqui specimen. (**B**)—Measurement model of the areas corresponding to each zone: TOA—(green area); TUA—(dark blue area); THU—(pink area); THS—(light blue area); standard length (yellow dotted line) and head length (red dotted line). (**C**)—Intermuscular position for counting. (**D**)—Epipleural intermuscular bones. (**E**)—Epineural intermuscular bones.

### 2.3. Dissection

The samples were chosen individually and randomly from the storage container and identified with the PIT-tag reader (Transponder Reader AT01, AnimaIITAG®, São Carlos—SP, BR). The PIT-tag number was noted on adhesive paper and fixed on the plastic plate, and it was also noted on the rectangular sheet of paper with the respective divisions of left and right sides. Then the sample was wrapped with aluminum foil, placed in a bain-marie machine (Banho Maria SL—150 Solab®, Piracicaba—SP, Brazil) that was kept at 80 °C and cooked for 40 min. At the end of this process, the sample was placed on the large plastic board in the right lateral decubitus position and dissected on the left side in the cranio-caudal direction.

Firstly, external cleaning was performed with the scalpel until the musculature could be visualized, and then dissection was started with the anatomical mouse tooth forceps in the following order of four anatomical zones: first zone, TOA (epaxial trunk—loin)—dorsal region, from the 3rd to the 14th vertebrae, where the last rib articulates; second zone, TUA (hypaxial trunk—ribs)—ventral region (ribs), from the 3rd to the 14th vertebra; third zone: THU (hypaxial tail—lower back)—ventral region, from the 14th to the last vertebra; fourth zone: THS (epaxial tail—upper back)—dorsal region, from the 14th to the last vertebra.

With this, all IBs from both fish sides were collected from the sarcomeres and organized in the same plastic plate in a sequence of collection in their respective zones, and an image was captured by a camera in a tripod (Canon T3i Canon® Professional Camera, Tokyo, Japan). Then the bone structures were transferred to the rectangular paper board, keeping them in the same positions, and fixed with adhesive tape and stored (Figure 2A).

### 2.4. Data Analysis

For the comparative analysis of both methods, each specimen was divided into the four anatomical zones: TOA, TUA, THS and THU (Figure 1B,C).

Using the PhotoScape v3.7 editing program, each X-ray image was enhanced for brightness, contrast and exposure to facilitate the visualization of small structures and morphological variations while preserving the structural features, as seen in Figure 1D,E.

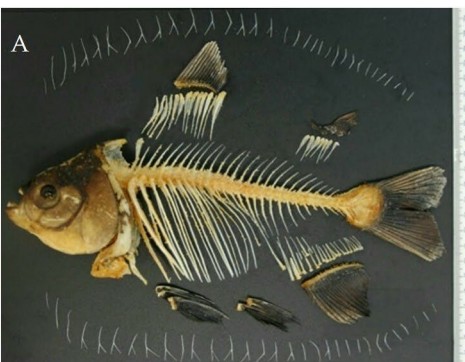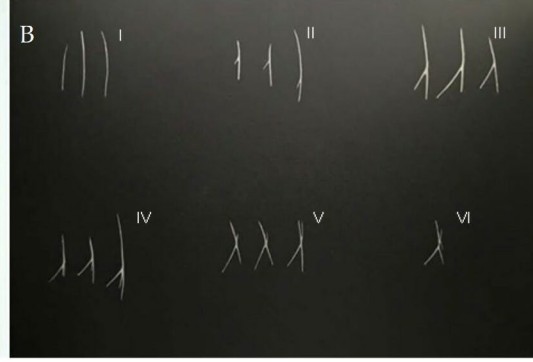

**Figure 2.** (**A**)—Completely dissected specimen with the different small bones organized according to the order of collection. (**B**)—Intermuscular bone types (type I = spine without forked; type II = incomplete forked spine; type III = Y-shaped spine; type IV = two-pronged spine type at one end; type V = two-pronged spine type at two ends; type VI = multiple forked spines at both ends).

The computer that was used to process these images was standardized at 50% screen brightness to avoid interference with edited parameter values. We opted for manual editing instead of applying filters so that this editing process could be done in any image editing program.

Selected quantitative and morphometric variables were analysed using ImageJ 1.51 j8 (National Institutes of Health, MA, USA; Java 1.8.0_112). The total number of intermuscular bones per specimen (left side) and numbers of bones per zone (TOA, TUA, THU, THS) were tabulated for both methods. IB morphology was classified into seven types following a previous study [32]: type I-I (spine without crotch), type II- ⊦ (incomplete crotch spine), type III-Y (Y crotch spine), type IV-OE (two-forked spine at one end), type V-TE (two-pronged spine), type VI-TM (two-pronged multiple-forked spine) and type VII-TB (three-branched spine). Each type of typology can be seen in Figure 2B, except for typology VII-TB. During statistical analysis, typology data from dissected material and the X-ray were identified using Roman numerals: bone typology by zone; bone length by zones (software: ImageJ1); areas of zones and head length/HL—from the tip of the snout to the 3rd vertebra. The following measurements were performed on X-ray images: standard length/SL—from the tip of the snout to the last vertebrae); areas of TOA, TUA, THU and THS and total area (sum of TOA, TUA, THU and THS). Statistical analyses were conducted in R v. 4.2.0 [33]. The above parameters from the fish body, body weight and IB measurements were expressed as the means ± standard deviation (SD), minimal and maximal observations. Next, data distribution was verified to analyze normality (Figure S1). We performed the Shapiro–Wilks test of the residuals for IB measurements after the Box–Cox approach to transform non-normal variables into normal shape. Then the strength of the relationships between dissections and X-ray methods, body weight and standard length were tested using Pearson's correlation. Four best predictive linear regression models were compared to estimate effects of standard length, body weight, TOA, THS, total area and multiple variables on intermuscular bone length in *C. macropomum*. Significance was accepted at the level of $p < 0.05$.

## 3. Results

### 3.1. Variations of Intermuscular Bones in Colossoma macropomum from Radiograph and Dissection Approach

In tambaqui, most of the epineural IB were the simplest type, having mainly two of three types, the type I-I and II- ⊦ (Table 1). The total number and length of IBs from X-ray imaging were strongly consistent with those obtained using the direct dissection (Table 2). The mean number of IBs was 26.976 (±6.03) using X-ray and 27.36 (±5.7) using the dissection method. The maximum number of IBs was 37 and 36, and the minimum number was 7 and 4 using X-ray and dissection, respectively. The mean value of the IB length was

6.16 mm (±1.88) using X-ray and 9.36 mm (±1.63) using the dissection method. The maximum IB length was 13.86 mm and 15.42 mm, and the minimum length was 2.86 mm and 5.21 mm using X-ray imaging and dissection, respectively. The differences between the two counting methods ranged from zero (no difference) to three IBs in 76.4% of fish samples analysed (Figure 3). The general correlation between X-ray radiography and dissection methods was moderately consistent with IB number (R Pearson = 0.82, *p* < 0.001) and IB length (R Pearson = 0.69, *p* < 0.001) (Figure 4A,B).

**Table 1.** Distribution of intermuscular bones by X-ray imaging method.

| Variable | Mean | SD | Min | Max | SE |
|---|---|---|---|---|---|
| Type I-I | 22.520 | 5.753 | 6 | 32 | 0.51 |
| Type II- � | 3.016 | 4.008 | 0 | 17 | 0.356 |
| Type III-Y | 1.449 | 1.602 | 0 | 9 | 0.142 |

**Table 2.** Intermuscular bones variation assessed via X-ray and dissection methods and standard length and body weight in tambaqui.

| Variable | Mean | SD | Min | Max | SE |
|---|---|---|---|---|---|
| Body Weight (g) | 228.862 | 78.931 | 92.700 | 457.500 | 7.117 |
| IBs Length by Dissection (mm) | 9.396 | 1.636 | 5.218 | 15.422 | 0.145 |
| IBs Length by X-ray (mm) | 6.167 | 1.880 | 2.860 | 13.869 | 0.167 |
| Standard Length (mm) | 185.471 | 22.135 | 142.680 | 238.250 | 1.964 |
| Tail Hindquarters Shaft (mm$^2$)—THS | 772.813 | 183.618 | 64.067 | 1191.050 | 16.294 |
| Tail Hindquarters Under Shaft (mm$^2$)—THU | 1715.700 | 410.774 | 949.071 | 2718.651 | 36.450 |
| Total Area (mm$^2$) | 8170.548 | 1891.288 | 4679.412 | 12,591.210 | 167.825 |
| Total Number of IBs by dissection | 27.362 | 5.700 | 4.000 | 36.000 | 0.506 |
| Total Number of IBs by X-ray | 26.976 | 6.031 | 7.000 | 37.000 | 0.535 |
| Trunk Over Axis (mm$^2$)—TOA | 2119.292 | 496.044 | 1142.359 | 3275.996 | 44.017 |
| Trunk Under Axis (mm$^2$)—TUA | 3562.743 | 835.990 | 2063.433 | 5618.769 | 74.182 |

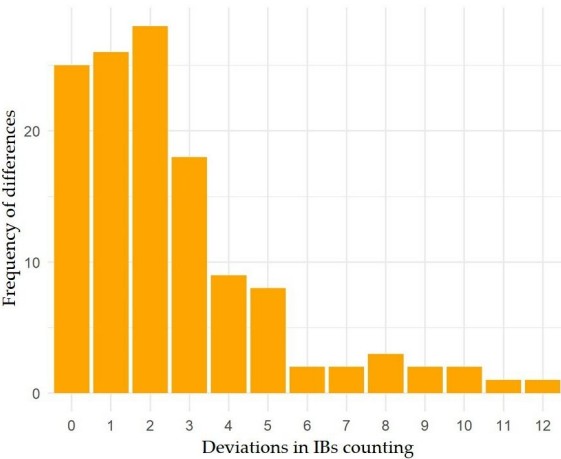

**Figure 3.** Frequency distribution of the deviation values in IBs numbers between X-ray imaging and dissection methods in tambaqui.

### 3.2. Predictive Models of IBs Based on X-ray Imaging in Tambaqui

We have found a moderate positive correlation between the total number of IBs by X-ray imaging and dissection methods for overall morphometric characters, varying from 0.48 (*p* < 0.001) to 0.54 (*p* < 0.001). Strong and positive correlations were demonstrated between IBs length by X-ray and zone areas (THU, THS and TOA; R > 0.70, *p* < 0.001), and there was also a moderate relationship to body weight and standard length (R = 0.68 and 0.69, *p* < 0.001) (Figure 5).

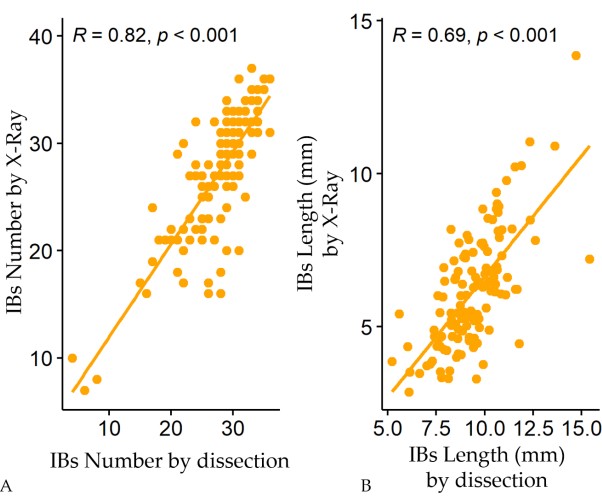

A

B

**Figure 4.** The relationship between tambaqui (**A**) IB number and (**B**) length in dissection and X-ray imaging methods (R Pearson, *p* < 0.001).

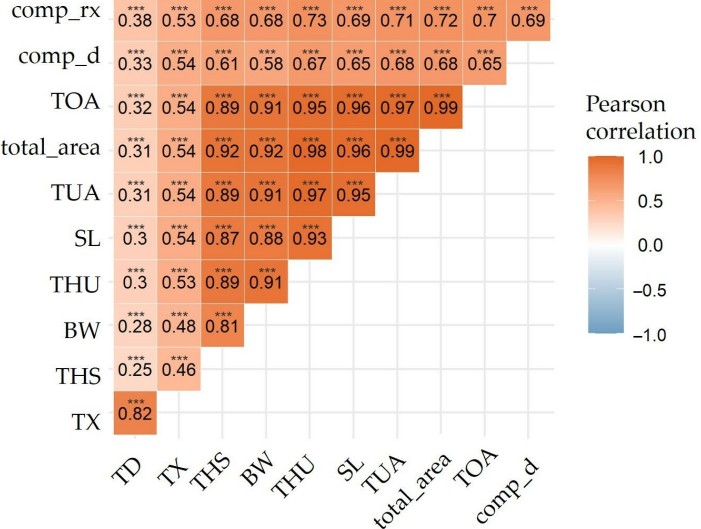

**Figure 5.** Pearson correlations of IB numbers and lengths from dissection and X-ray imaging methods against morphometrical parameters in tambaqui. TD = total number of IBs by dissection, TX = total number of IBs by X-ray, comp_d = bones length by dissection (mm), comp_rx = bones length by X-ray (mm), SL = standard length (mm), BW = body weight (g), total_area = total area of lateral loin (mm$^2$), TOA = trunk over axis (mm$^2$), TUA = trunk under axis (mm$^2$) THU = tail hindquarters under shaft, THS = tail hindquarters shaft (mm$^2$). *** *p* < 0.001. The Shapiro–Wilks test of the residuals for IB length after Box–Cox transformation indicated that the data were normally distributed (W > 0.99362, *p*-value > 0.8527). Although the histogram of the IB number residuals was approximately symmetrical, there were a few outliers in the normal QQ plot, and the Shapiro–Wilks test indicated a deviation from normality (W > 0.9551; *p* < 0.001). However, the plot of the residuals versus predicted values indicated that they were independent, and the IB lengths predicted by the model and the measured lengths and body weight varied linearly.

The IB length from the X-ray imaging method in tambaqui was influenced by standard length 0.01 (95% CI 0.01, 0.01; *p* < 0.001) (Figure 6A), body weight 0.006 (95% CI 0.00, 0.00; *p* < 0.001) (Figure 6B) and trunk over axis 0.04 (95% CI 0.00, 0.00; *p* < 0.001) (Figure 6C). Body weight and TOA also affected IB length in tambaqui during grow-out, but to a lesser degree.

There was no significant interaction or synchronic effect on IB length in tambaqui during growth performance when considering the predictors of standard length, body weight, TOA and THS together 0.00 (95% CI 0.00, 0.01; *p* = 0.7) (Figure 6D).

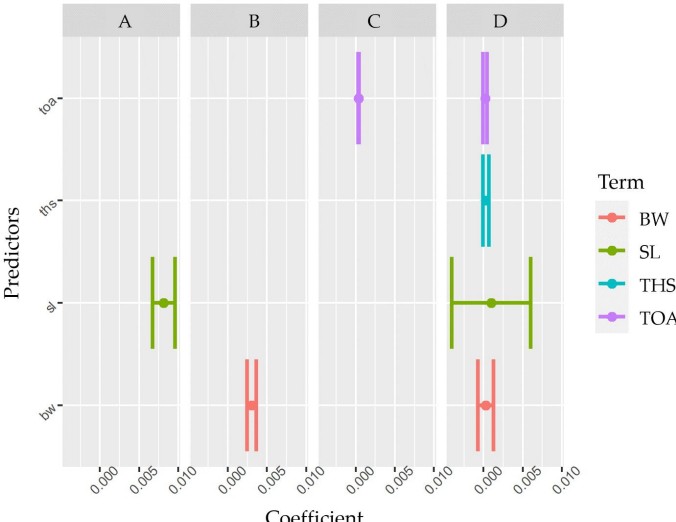

**Figure 6.** Predicted models for intermuscular bones length in tambaqui. Estimation of effects on intermuscular bones length in tambaqui (*Colossoma macropomum*). (**A**)—standard length effect model. (**B**)—body weight effect model. (**C**)—trunk over axis effect model. (**D**)—multiple effect model.

The regression analysis of IBs (y) and growth parameters (x) showed that they fit an optimal model considering standard length ($R^2$ = 0.50, *p* < 0.0001), which indicated a positive correlation between IB length from X-ray and standard length (Figure 7A). The same pattern was found between IB length and body weight ($R^2$ = 0.46, *p* < 0.0001), TOA ($R^2$ = 0.53, *p* < 0.0001), THS ($R^2$ = 0.48, *p* < 0.0001) and total area ($R^2$ = 0.54, *p* < 0.0001; Figure 7B–E).

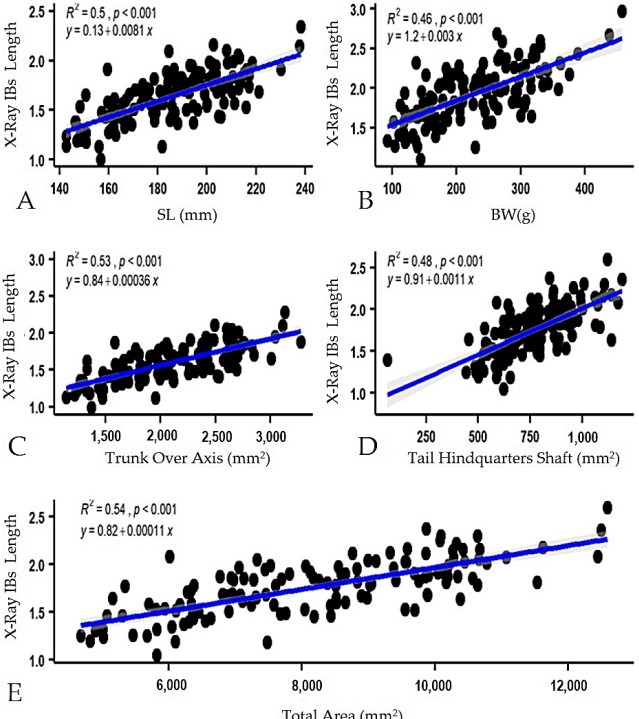

**Figure 7.** Effects of growth parameters in: (**A**) SL (mm); (**B**) BW (g); (**C**) trunk over axis (mm$^2$); (**D**) tail hindquarters shaft (mm$^2$); (**E**) total area (mm$^2$) on IB length from X-ray imaging in tambaqui *Colossoma macropomum*. Regression lines are calculated from coefficients from linear model, after Box–Cox transformation for normalization of data and 95% confidence level of fitted values. $R^2$ are the coefficients of determination adjusted accounting for specific effects.

## 4. Discussion

Our study is the first quantitative study involving IB length variations associated with parameters of fish growth in *C. macropomum*. By comparing IB morphological characteristics using the direct dissection method and high-resolution images from X-ray imaging, we have demonstrated that X-ray imaging is a better non-lethal method to investigate IB variation in tambaqui and that correlation analysis and simple linear analysis of quantitative models can predict IB lengths and position in tambaqui. The existence of IBs in tambaqui has an extremely negative impact on their edible and economic value. Thus, IBs-free fishes would be of enormous significance in both basic research and aquaculture. After the findings of natural specimens of tambaqui lacking intermuscular bones [27], research has revealed the genes associated with the absence of intermuscular bones [30]; as such, the genetic and molecular mechanisms underlying the expression of such desirable phenotypes are currently better known [34]. Current genome-editing techniques, such as CRISPR-Cas9 [35], can be developed to generate new IB-defected strains; however, methods such as X-ray imaging can be essential, complementary and integrative in deducing and describing the IB morphology [18,20,36]. Between these two methods, X-ray imaging to count and measure IBs is a rapid and non-invasive technique and has thus recently become a very convenient and popular approach for various fish species [26,27,37]. Comparing different methods in reference samples is crucial to obtain associative relationships between variables that are difficult to measure without killing the fish. This will lead to finding more efficient and cheaper data collection methods, as in our study. Our results show that the IB morphological characteristics from X-ray images were not significantly different from those from the dissection method (Figure 3), and both methods showed a strong correlation between them (Figures 4 and 5). The dissection method was very laborious and time-consuming, and most importantly, the specimen needed to be killed. While superior due to ease of use and accuracy, the drawback of an X-ray machine is the availability. For breeding programs, collection of desired trait data by non-lethal means is important, and use of X-ray imaging satisfies the collection of IB data in tambaqui with accuracy.

Our results show that the size of intermuscular bones had a strong correlation with morphometric measurements (body weight and standard length), meaning IB length becomes larger as the specimen grows. The number of intermuscular bones had a moderate correlation to body weight and standard length (Figure 5). Our study was limited by the size of fish available for X-ray and dissection investigation, and larger fish (>23.83 cm SL) could not be included. To the best of our knowledge, this information (number of IBs in adult tambaqui and correlation to body weight and standard length) is not available in the literature for any fish species. In our study, there was a weak correlation between fish length and IB number within the fish sizes we used, but extrapolation of this result to IB number in larger animals may not be possible and may be inaccurate. In *C. macropomum*, the number of IBs is determined early in the larval stages, and a complete IB formation is reported in individuals of 2.6 cm in total length [38]. Therefore, within the fish size in the studied material in our study, individuals should already have a definite IB number, so the correlation we found needs to be interpreted more carefully. We believe that since IB ossification increases with fish growth, smaller fish that were analysed in our study would still have uncalcified tendons [39] that could not have been depicted either through X-ray or dissection methods in smaller individuals. Depicting these tendons would require other techniques (i.e., histology) and is out of the scope of this study. In larger and older tambaqui, IBs are expected to be more calcified to support their own weight and mobility needs, so the X-ray technique would become even more useful (considering IB density, contrast, sharpness and magnification parameters of X-ray operation) and could facilitate observation of IBs in larger animals.

To date, the most common methods used to study the IBs in fish are histology using staining, anatomy using dissection and radiography using X-ray or ultrasound [40]. While histology using staining is useful for studying the development of IB in the early larval and juvenile stages, it would be difficult to obtain a precise count of IBs [23]. The use of X-rays or ultrasound is relatively simple and can be used to count intermuscular bones non-invasively,

but the equipment is expensive. The anatomy method can provide a precise IB count but is lethal and labour-intensive. Like ours, a few studies have used anatomy, histology and X-ray imaging to examine the IB morphology and number [26,40–42]. Yang, Jiang, Wang, Zhang, Pan and Yang [40] reported similar numbers of IBs using both anatomical and X-ray imaging methods in different cyprinid sub-families, as in our study on tambaqui. Although other studies used both methods to examine the IB number and morphology, they only reported the results from the X-ray, indicating this method alone can provide accurate information on IB number and morphology [26,41,42].

At the species level, numbers of IBs in fish are relatively stable for several carp species studied [22], but significant intra-specific variations have been found in species such as the blunt snout bream *Megalobrama amblycephala,* whose IB number varied between 84 and 146 [43]. Comparing different species, the number, length and morphology of IBs were found to vary depending on the environmental factors, types of diets [40], different swimming modes [19], events of species hybridizations [23,41] and the phylogenetic history that shaped life history patterns in these fish species [44,45]. In Cyprinidae, for example, IB numbers can range from 73 to 169 [22,44], with higher numbers in carnivorous species compared to herbivorous species [40]. When comparing the number of IBs in fish on different sides (left vs. right) of the same species, statistical differences appear to be insignificant [23,26]. In our study, we found 26.97 ($\pm$6.0) IBs in the left side of the fish and a sum of 53.9 IBs in both sides (counted from dissected fish). Total IB number in tambaqui is lower than counts made for other omnivorous species, such as several carp species and zebrafish [23,44]. Although fewer in number, IBs in tambaqui were found to be highly variable in terms of different types (Table 1). The hypaxial tail (lower back)—THU was found to have a reduced number and length of IBs with high variation (see Supplementary Table S1), which suggests that the THU might be an optimal target for selective breeding for fewer IBs in tambaqui. Although the TOA has the higher number of IBs, it is the part of the tambaqui loin most appreciated by the customers. While THU has the potential for selection due to the reduced IBs in this region, due to its preference among consumers, TOA could still be a target for selective breeding to reduce the IBs. Several molecular techniques (microsatellites, SPNs) have been developed in the last couple of decades [18,43,46,47] which can be successfully applied in selective breeding for desirable traits. A recent study [43] of snout bream (*Megalobrama amblycephala*) found a significant additive genetic variation for IB number and concluded that moderate heritability in IB number in snout bream can be considered as a trait for selection in selective breeding programs.

Some internal structures of fish increase as they grow. Tambaqui, a rounded fish, has a large loin region of high muscular density where the myosepts and intermuscular bones are accommodated [27]. The number and length of myosepts reflect, to some extent, the IBs number and their length, associated with vertebrae and other morphogeometrical shapes [40]. Results from our study demonstrate a strong association between morphometrical structures (Figure 5. TOA, TUA, SL, BW) and IB length, which confirms the close relationship between the muscle mass and morphogeometry of the tambaqui filet and its intermuscular bone structures. The use of a single and independent predictor variable seems to be the best way to predict IB length in tambaqui. The single effect of standard length, body weight and TOA provided the best predicted models in our study (Figures 6 and 7). From this simple predictive model, it is possible to make a preliminary estimation of IB length by extrapolating from the standard length, body weight and TOA of tambaqui.

## 5. Conclusions

In our study, we showed that both the dissection and X-ray imaging methods provided similar intermuscular bone counts and morphologies in tambaqui and that X-ray methods can replace the labour-intensive and time-consuming dissection method. Further, because the tambaqui stakeholders are aiming to develop a selective breeding program for tambaqui and prefer to use the variation in intermuscular bone count as a trait in the selective breeding program (aiming to reduce or eliminate), using a non-lethal method to

characterize the IBs in tambaqui will be seen as a positive development. In consideration of the many advantages of X-rays, such as ease of operation, timesaving qualities, no damage to specimens and especially applicability to live fishes, we recommend using X-ray imaging for intermuscular bone validation in tambaqui.

**Supplementary Materials:** The following supporting information can be downloaded at: https://www.mdpi.com/article/10.3390/fishes8040180/s1, Table S1—Intermuscular bone and morphometrical variation by zones in Colossoma macropomum; Figure S1—Histograms of all variables related to IBs in tambaqui from the present study.

**Author Contributions:** Conceptualization, E.S.V. and V.P.; methodology, E.S.V., L.K.F.d.L., V.P., T.H.D., J.J.T.d.S., A.d.S.B. and N.F.A.O.; software, N.F.A.O. and E.S.V.; validation, N.F.A.O. and E.S.V.; formal analysis, N.F.A.O. and E.S.V.; investigation, N.F.A.O. and E.S.V.; resources, E.S.V. and L.K.F.d.L.; data curation, N.F.A.O. and E.S.V.; writing—original draft preparation, N.F.A.O.; writing—review and editing, E.S.V., L.S.T., V.P., L.A.B.-C., T.H.D., J.J.T.d.S and A.d.S.B.; visualization, E.S.V.; supervision, L.A.B.-C., E.S.V. and L.K.F.d.L.; project administration, ESV and L.S.T.; funding acquisition, E.S.V. and L.S.T. All authors have read and agreed to the published version of the manuscript.

**Funding:** This research was funded by the European Union's horizon 2020 research and innovation programme under grant agreement N° 818173. N.F.A.O. was funded by Coordenação de Aperfeiçoamento de pessoal de nível superior—Brasil (CAPES)—Finance Code N °001.

**Institutional Review Board Statement:** The study was conducted in accordance with the Brazilian guidelines for the care and use of animals for scientific and educational purposes–DBCA and was approved by the National System for the Management of Genetic Heritage and Associated Traditional Knowledge—SISGEN (ADBE614), as well as by the Ethics Committee for the Use of Animals—CEUA of the National Research Centre on Fisheries, Aquaculture and Agricultural Systems—CNPASA (specific protocol N° 11/2018).

**Data Availability Statement:** Data supporting the reported results can be found in https://doi.org/10.5281/zenodo.7729757.

**Acknowledgments:** The authors are grateful to "Fazenda Aquicultura São Paulo" (Brejinho de Nazaré-TO, Brazil) for assistance in fish sampling during the work, to "UniCatólica-Centro Universitário Católica do Tocantins" (Palmas-TO) for allowing them access to X-ray equipment and to Ronivaldo Bento de Souza for X-ray machine operation. Thanks also to those colleagues who helped with fish sampling: Aristóteles Gomes Caponi and Ramon Lacerda Maciel. We thank three anonymous reviewers for their helpful comments and Nivetha Puvanendran for the grammatical editing on this manuscript.

**Conflicts of Interest:** The authors declare no conflict of interest. The funders had no role in the design of the study; in the collection, analyses, or interpretation of data; in the writing of the manuscript or in the decision to publish the results.

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
