# Peer review of "Fish Size Correlates to Size and Morphology of Intermuscular Bones in Tambaqui Colossoma macropomum as Shown by Dissection and X-ray Imaging Methods"

_fishes, doi:10.3390/fishes8040180_

Round 1
Reviewer 1 Report
Dear Authors,
The present study is devoted to an actual topic - methods for assessing bone structures in tambaqui (Colossoma macropomum) - one of the six most popular species for aquaculture in Brazil. The processing of this fish after cultivation and obtaining high-quality fillets are significantly complicated by presence of intermuscular bones, and reducing their number in farmed fish through selection breeding is an extremely important issue. Therefore, the development of in vivo methods for skeletal studies in cultivated tambaqui, which is the aim of the study, is considered as an important direction for studies and progress in fish aquaculture in Brazil.
In the course of the study, two methods for intermuscular bones analysis were compared, - traditional, with use of heat treatment and subsequent skeletal analysis, and more modern, with use of X-rays. The work confidently shows that both methods give similar results, which allowed the authors to reasonably conclude that the X-ray method can be an effective approach to the intravital study of intermuscular bones in artificially cultivated tambaqui. In general, the study and manuscript makes a good impression, is conducted on a representative material with use of adequate methods, includes a discussion and conclusion based on the results, therefore it is recommended for publication with minimal changes.
Some inaccuracies were found in the text of the manuscript, and recommendations for their correction were given, noted below. I believe that the correction of these edits will improve the manuscript, and will not be a significant effort for the authors.
Best regards
General comments
Though the study was conducted on quite limited size range of young individuals (14-24 cm, 8 mo old), authors found significant correlates between fish size and number and morphology of intermuscular bones. Does it mean that this relationship can be seen in older/larger fish too? If authors can see more bones in larger fish, is it a result of development of new intermuscular bones in older fish, or it’s a result of their easier availability for analysis? Larger fish can be more difficult for taking x-ray pictures for analysis and, hence, less available for x-raying. It would be interesting to find some ideas about this in the Discussion section. Another interesting point is development of intermuscular bones during fish ontogenesis. When number of these structures is finally determined in fish, and what is a common size of cultivated tambaqui when harvested? Is it possible that new intermuscular bones develop in old tambaqui? It would be great if authors could add some ideas on these topics in the Discussion section, maybe with reference to other studies.
Specific comments
Line 172. “Standard length/SL – from the tip of the snout to the caudal peduncle”. It’s not quite clear from this description what was the most posterior point for SL estimation. Should not the “caudal peduncle” be changed to the “last vertebra”?
Line 184. “B - Intermuscular bone models” should be changed to “B - Intermuscular bone types” or “B - Intermuscular bone shapes”, as for me.
Lines 205-206. Table 2.
This table includes names of characters and it looks quite illogical that the two trunk characters have too different names if compared with the two tail characters ((1) - Trunk Over Axis - TOA, (2) - Trunk Under Axis – TUA, (3) - Tail Hindquarters Under Shaft – THU, (4) - Tail Hindquarters Shaft – THS). I’m not sure if that characters (3) and (4) are called righty, although I understand that it was a way of getting different abbreviations for them. Maybe other names can be used? For example, anterior and posterior, dorsal and ventral body parts. In this case, all abbreviations will be different (like AD, AV, PD and PV).
Line 207. Fig. 3. There is a misprint in legend for x-axis: “countig” should be changed to “counting”.
Line 252. Fig. 7 demonstrates effects of growth parameters on “IB length and IB number” (line 252). Can it be specified which figures demonstrate effects on IB length, and where are effects on IB number? It is not shown in legends for y-axis.
Lines 257-258. “Our study is the first quantitative study involving IB length and number variations associated to morphometrical growth phenotypes in C. macropomum”. It’s not quite clear what is a meaning of phrase “morphometrical growth phenotypes”. Which growth phenotypes were analyzed? Could it be rephrased somehow? Do you mean “parameters of fish growth”?
Line 301. There is a reference to the Table S2, though there is no reference to the Table S1 in the text. Is it a misprint? The same is applicable to the Table S2 per se (should be changed to S1?).
Reviewer 2 Report
This manuscript “Fish size correlates to number and morphology of intermuscular bones in tambaqui Colossoma macropomum as shown by dissection and X-ray imaging methods” presented quantitatively examined the intermuscular bones variation on farmed tambaqui by comparing the direct anatomical dissection with the high-resolution X-ray imaging method. This work demonstrated that the X-ray method can provide accurate phenotypic data (in vivo) for IB counting and length measurements by extrapolating from the standard length, body weight, and trunk over-axis area of tambaqui. Statistical analysis was also conducted to prove the experimental data were statistically significant. The experimental procedures are sound, and the objectives are achieved. However, the relevance of the study is not clear. Moreover, I did not see the authors analyze the data in detail and explained how the data values can lead to conclusions. Therefore, the manuscript especially the introduction and discussion parts should be significantly amended by the authors. Here are some more detailed comments:
This manuscript has minor spelling and grammatical errors and would benefit from closer proofreading.
Introduction section. Seems that the study will be relevant only to Brazil and to only one fish species. Moreover, it is not clear why to start with the nutritional value of fish, if the problem is the tambaqui marketing, due to intramuscular “Y” bones. Please clarify the following questions:
Was observed this characteristic in other fish species?
Why high-resolution X-ray imaging methods were used for the diagnosis and quantification of the tambaqui IBs?
Why compare with dissection?
Recommendation: please rewrite and add references to the new information
Figure 7. Why use the term growth parameters? Accordingly, with the methodology, the study analyzed only juveniles and did not make measurements over different periods of time. Please clarify this point.
Discussion section: Can you discuss the X-ray results compared with other studies? Can you explain the difference between your proposal methods and others? I would like to see references here of Why X‐rays are considered advantageous and compare it with results in similar studies?
The conclusion is not corresponding to the results. Please rewrite.
Reviewer 3 Report
The article "Fish size correlates to number and morphology of intermuscular two bones in tambaqui Colossoma macropomum as shown by three dissection and X-ray imaging methods" shows the first characterization of the number and morphology of intermuscular bones in a neotropical fish. As a study case report, this new information is worth to be published. However, the manuscript needs some major revisions before publishing.
Introduction
I suggest restructuring the introduction section. The study aims to characterize the intermuscular bones in fish, particularly in neotropical fish. As far as I'm concerned, no such analysis of neotropical fish exists. On the other hand, there are plenty of Asia fish species. The authors focus the introduction on aquaculture. Although the intermuscular bones are a challenging phenotype for aquaculture, this could just be mentioned and highlight what these bones are for the readers. Some reviews have been published on the subject, which the authors could use and cite in the introduction.
Nie, C.H., Hilsdorf, A.W., Wan, S.M. and Gao, Z.X., 2020. Understanding the development of intermuscular bones in teleost: status and future directions for aquaculture. Reviews in Aquaculture, 12(2), pp.759-772. (cited in the discussion, however, about the X-ray use).
Patterson C, Johnson GD (1995) The intermuscular bones and ligaments of teleostean fishes. Smithsonian Contribution to Zoology 559: 1–85.
Mubango, E., Tavakoli, S., Liu, Y., Zheng, Y., Huang, X., Benjakul, S., Yuqing, T., Luo, Y. and Hong, H., 2022. Intermuscular Bones in Asian Carps: Health Threats, Solutions, and Future Directions. Reviews in Fisheries Science & Aquaculture, pp.1-26.
About Colossoma macropomum, I've found a review worth to be cited, too.
Hilsdorf, A.W.S., Hallerman, E., Valladão, G.M.R., Zaminhan‐Hassemer, M., Hashimoto, D.T., Dairiki, J.K., Takahashi, L.S., Albergaria, F.C., Gomes, M.E.D.S., Venturieri, R.L.L. and Moreira, R.G., 2022. The farming and husbandry of Colossoma macropomum: From Amazonian waters to sustainable production. Reviews in Aquaculture, 14(2), pp.993-1027.
Line 42: Grouper is not an important fish-farmed fish species in Brazil. That's for sure.
Lines 59-62: The sentence's content doesn't match the reference. The reference is a Letter to the Editor of Epilepsy & Behavior journal about fatty acids found in tambaqui. The authors of this Letter state, "In conclusion, researchers clearly demonstrated that the tambaqui is a rich source of omega-3 FAs, and this fact should be disclosed and, mainly, consumed by the general population". They don't say about high acceptance. In the tambaqui's review, the authors can find this information.
I couldn't find any citation about other studies of intermuscular bones in fish, which can be found in many articles published by Chinese and Indian authors.
In short, introduction must be rewrite so that the reader can understand better the importance of the article´s aims and why the information is relevant to be published.
Methods
Firstly, the authors must provide the study's Animal Ethics Committees (AECs).
Lines 132-136: The authors could present the four anatomical zones in a pictorial figure so that the reader can understand better the areas where the IB are localized.
Line 144: Figure 1 photos need to be enlarged so that the reader can better visualize the X-ray details
Discussion
The authors assert that X-ray is a better non-lethal methodology to quantify IB number in tambaqui. Maybe it's not the only one. Perazza et al. (2017) use ultrasound to verify the presence and number of IB in tambaqui. This methodology is also non-lethal and may be more practical in fieldwork. The authors could mention this other methodology.
Concerning IB morphology, despite the authors showing a Person correlation between IB variables (b=number and length), the X-ray images can only distinguish the different types of IB without dissection.
Line 302-304: The authors state, "While THU has the potential for 303 selection due to the reduced IBs in this region, TOA would still be a target for selective 304 breeding to reduce the IBs". What do the authors mean? Is there any indication that the number reduction of IB can be selected? There are studies with carp showing that was not possible. The molecular and genetic mechanism underlying this trait must first be known before including it in a selective breeding program.
Moav, R., Finkel, A. and Wohlfarth, G., 1975. Variability of intermuscular bones, vertebrae, ribs, dorsal fin rays and skeletal disorders in the common carp. Theoretical and Applied Genetics, 46(1), pp.33-43.
Li, L., Zhong, Z., Zeng, M., Liu, S., Zhou, Y.I., Xiao, J., Wang, J. and Liu, Y., 2013. Comparative analysis of intermuscular bones in fish of different ploidies. Science China Life Sciences, 56, pp.341-350.
Round 2
Reviewer 2 Report
Introduction
P1, L36-40. The interest of the readers in the per capita fish consumption in Brazil could be very low. Suggestion, erase, or rephrased this phrase.
P1, L41-43. Should be more interesting to mention the main farmed species worldwide and can include tambaqui as specific specie in Brazil. Suggestion rephrased this phrase.
P2, L64-69. Rephrased this paragraph, it is low relevant or attractive the following information “In North Brazil, the tambaqui is considered a noble fish compared to most of the 64 commercialized species of the highly diversified Amazonian region. Its high acceptance 65 by the consumers is related to the taste, attractive white colour and presence of reasonable 66 fat and high protein content of the meat..”
P12, L380-389. Conclusions: Suggestion, do not use IB, replace with intermuscular bones
Author Response
see the uploaded file

Reviewer 3 Report
I still have two additional issues to address Firstly, despite Valenti's article's citation about grouper production in Brazil. I insist there is no relevant on-growing dusky grouper production in Brazil. If the authors check it out in the IBGE data or other sources, you don't see any figures about farmed dusky grouper in Brazil. I advise removing this and leaving just the main farmed species. Regarding lines 360-364. I suggest: "While THU has the potential for selection due to the reduced IBs in this region, TOA would still be a target for selective breeding to reduce the IBs. Despite this, a recent study [45] in snout bream (Megalobrama amblycephala) found a significant additive genetic variation for IB number. It concluded that moderate heritability in IB number in snout bream is still challenging since this approach's first works have failed (Meske, 1968). Meske, C., 1968. Breeding carp for reduced number of intermuscular bones and growth of carp in aquaria. Bamidgeh, Bulletin of Fish Culture in Israel, 20(4), pp.105-119.
Author Response
see the uploaded file
